# Hydra: A mixture modeling framework for subtyping pediatric cancer cohorts using multimodal gene expression signatures

**Jacob Pfeil**[1,2]*, **Lauren M. Sanders**[1,2,3], **Ioannis Anastopoulos**[1,2], **A. Geoffrey Lyle**[2,3], **Alana S. Weinstein**[1,2], **Yuanqing Xue**[1,2], **Andrew Blair**[1,2], **Holly C. Beale**[2,3], **Alex Lee**[4], **Stanley G. Leung**[4], **Phuong T. Dinh**[4], **Avanthi Tayi Shah**[4], **Marcus R. Breese**[4], **W. Patrick Devine**[5], **Isabel Bjork**[2], **Sofie R. Salama**[1,2,6‡], **E. Alejandro Sweet-Cordero**[4‡], **David Haussler**[1,2,6‡], **Olena Morozova Vaske**[2,3‡]

**1** Department of Biomolecular Engineering, University of California, Santa Cruz, Santa Cruz, California, United States of America, **2** Genomics Institute, University of California, Santa Cruz, Santa Cruz, California, United States of America, **3** Department of Molecular, Cell and Developmental Biology, University of California, Santa Cruz, Santa Cruz, California, United States of America, **4** Department of Pediatrics, Division of Hematology and Oncology, University of California, San Francisco, San Francisco, California, United States of America, **5** Department of Anatomic Pathology, University of California, San Francisco, California, San Francisco, United States of America, **6** Howard Hughes Medical Institute, University of California, Santa Cruz, Santa Cruz, California, United States of America

‡ Senior authorship.
* jpfeil@ucsc.edu

**Data Availability Statement:** All relevant data are within the paper and its Supporting Information files. The input data and analyses are available in an

## Abstract

Precision oncology has primarily relied on coding mutations as biomarkers of response to therapies. While transcriptome analysis can provide valuable information, incorporation into workflows has been difficult. For example, the relative rather than absolute gene expression level needs to be considered, requiring differential expression analysis across samples. However, expression programs related to the cell-of-origin and tumor microenvironment effects confound the search for cancer-specific expression changes. To address these challenges, we developed an unsupervised clustering approach for discovering differential pathway expression within cancer cohorts using gene expression measurements. The hydra approach uses a Dirichlet process mixture model to automatically detect multimodally distributed genes and expression signatures without the need for matched normal tissue. We demonstrate that the hydra approach is more sensitive than widely-used gene set enrichment approaches for detecting multimodal expression signatures. Application of the hydra analysis framework to small blue round cell tumors (including rhabdomyosarcoma, synovial sarcoma, neuroblastoma, Ewing sarcoma, and osteosarcoma) identified expression signatures associated with changes in the tumor microenvironment. The hydra approach also identified an association between *ATRX* deletions and elevated immune marker expression in high-risk neuroblastoma. Notably, hydra analysis of all small blue round cell tumors revealed similar subtypes, characterized by changes to infiltrating immune and stromal expression signatures.

open-source repository at https://github.com/jpfeil/hydra-paper.

**Funding:** We would like to thank Emily Beazley Kures for Kids Fund St. Baldrick's Consortium Grant (427053), California Initiative to Advance Precision Medicine (OPR014109), Alex's Lemonade Stand Foundation for Childhood Cancer Research (https://www.alexslemonade.org/), Unravel Pediatric Cancer (https://unravelpediatriccancer.org/), Team G Childhood Cancer Foundation (https://teamgfoundation.org/), Live for Others Foundation (http://l4of.org), and the National Human Genome Research Institute T32 Training Award (3T32HG008345-01S1). OMV holds the Colligan Presidential Chair in Pediatric Genomics. DH is a Howard Hughes Medical Institute Investigator. The funders had no role in study design, data collection and analysis, decision to publish, or preparation of the manuscript.

**Competing interests:** I have read the journal's policy and the authors of this manuscript have the following competing interests: Olena Vaske's spouse has stock interests in NantHealth.

## Author summary

Pediatric cancers generally have few somatic mutations. To increase the number of actionable treatment leads, precision pediatric oncology initiatives also analyze tumor gene expression patterns. However, currently available approaches for gene expression data analysis in the clinical setting often use arbitrary thresholds for assessing overexpression and assume gene expression is normally distributed. These methods also rely on reference distributions of related cancer types or normal samples for assessing expression distributions. Often adequate normal samples are not available, and comparing matched cancer cohorts without accounting for subtype expression overestimates the uncertainty in the analysis. We developed a computational framework to automatically detect multimodal expression distributions within well-defined disease populations. Our analysis of small blue round cell tumors (including rhabdomyosarcoma, synovial sarcoma, neuroblastoma, Ewing sarcoma and osteosarcoma) discovered a significant number of multimodally expressed genes. Multimodally expressed genes were associated with proliferative signaling, extracellular matrix organization, and immune signaling pathways across cancer types. Expression signatures correlated with differences in patient outcomes for *MYCN* non-amplified neuroblastoma, osteosarcoma, and synovial sarcoma. The low mutation rate in pediatric cancers has led some to suggest that pediatric cancers are less immunogenic. However, our analysis suggests that immune infiltration can be identified across small blue round cell tumors. Thus, further research into modulating immune cells for patient benefit may be warranted.

## Introduction

Large cancer sequencing projects, including The Cancer Genome Atlas (TCGA) and Therapeutically Applicable Research to Generate Effective Treatments (TARGET), have facilitated the development of cancer gene expression compendia [1–6], but these compendia often lack expression data from corresponding normal tissue. Without the normal comparator, Hoadley et al. (2018) found that cell-of-origin signals drive integrative clustering of TCGA data. Strong cell-of-origin and tumor microenvironment (TME) signals may also complicate the interpretation of gene expression results for precision oncology applications, so careful modeling of the data is necessary to infer accurate conclusions.

The TME includes tumor cells, stromal fibroblasts, immune cells, and vasculature [7]. Similarities in TME composition across tumor samples have led to the identification of TME states (e.g. inflamed, immune-excluded, immune-desert). While these states are dynamic, they can still shed light on the immunogenicity of tumor cells and correlate with response to cancer immunotherapies [8]. The TME cellular composition can be inferred from tumor RNA-Seq data since host cell RNA is sequenced along with the cancer cell RNA. Tumor progression and response to therapies is associated with features of the TME. Therefore, targeting the TME therapeutically may improve treatment outcomes in some cancers.

Immunotherapies that activate the host immune system to eradicate tumors have been effective in treating several cancer types, particularly cancers with a high mutation burden [9, 10]. Pediatric cancers tend to have fewer mutations than adult cancers, and while there has been limited testing of immunotherapies in pediatric cancer patients, the currently available data suggest lower response rates than adult cancers [11, 12]. However, improved immune subtyping of pediatric cancers may identify subsets of patients that are candidates for powerful immunotherapies. In addition to infiltrating immune cells, cancer-associated fibroblasts

(CAFs) assist in extracellular matrix remodeling and activation of growth factor signaling. CAFs facilitate tumor growth, metastasis, and resistance to some therapies, so identification of CAF functions within a tumor may also facilitate clinical decision making. Methods are needed to both infer and characterize gene expression subtypes that correlate with tumor microenvironment states to accelerate the development of personalized therapies for pediatric cancers.

Tumor/normal differential expression analysis in which a cohort of tumor tissues is compared to corresponding normal tissue samples is an effective approach for identifying gene expression biomarkers [13–15], but it is often not possible to conduct this analysis in a clinical setting. Sufficient biological and technical replicates are limited by tumor tissue availability, and healthy neighboring tissue often cannot be isolated. In addition, for many pediatric cancers, the cell-of-origin, and thus the appropriate reference normal tissue, is not known. Besides differential expression analysis, single-sample pathway analysis can be used to identify upregulation of biological gene sets in tumor subtypes. Among the most widely used pathway analysis approaches is gene set enrichment analysis (GSEA) [16, 17]. GSEA identifies coordinated expression of pathway genes using gene ranks and a Kolmogorov-Smirnov-like test statistic. GSEA is usually performed on differentially expressed genes to compare two cohorts or phenotypes, but single-sample GSEA is also available when there is not an obvious comparator. GSEA uses curated pathway gene sets like those in the Molecular Signatures Database (MSigDB) [18].

Cancer gene expression subtypes are traditionally identified using unsupervised clustering methods such as consensus clustering analysis [19–21]. These methods are generally underpowered because the number of genes greatly exceeds the number of samples. Dimensionality reduction approaches such as Principal Component Analysis (PCA) have been found to underestimate the dimensionality of gene expression data [22]. Lenz at al. (2016) found two cases in which PCA fails to identify a biological signal: when the size of the cluster is small and when the effect size is small. Lenz et al. (2016) suggests investigating multimodally expressed genes to improve identification of cancer subtypes. Cancer subtypes naturally lead to multimodal expression patterns because each subtype expresses a correlated set of genes at different expression levels. Expression subtypes may result from dysregulated pathway expression within cancer cells, but another source of multimodal expression comes from varying amounts of infiltrating immune and stromal cells in the TME.

Gaussian mixture models are a powerful class of unsupervised clustering algorithms that can be used to detect multimodally expressed genes [23–25]. A Gaussian mixture model is appropriate when the expression data can be modeled as a mixture of two or more Gaussian distributions [26]. One limitation of Gaussian mixture models in this context is that the number of clusters in the data is often not known beforehand, so a parameter search must be used to identify the best-performing model. However, this is a computationally expensive approach. This problem can be overcome by placing a Dirichlet process prior on the number of expression clusters. The number of clusters is then inferred while fitting the mixture model using Markov chain Monte Carlo (MCMC) sampling [26]. This approach has not been widely used in clinical cancer research because these algorithms are still computationally expensive, but recent advances in Bayesian variational inference have made this approach scalable for precision oncology applications [27].

Here, we present the hydra framework for identifying clinically relevant expression subtypes and classifying N-of-1 tumor samples using learned models. We provide an overview of the hydra framework, assess performance for detecting differential pathway expression, and apply the framework to better understand expression patterns in high-risk neuroblastoma and other small blue round cell tumors. We apply the learned models trained on publicly available

cancer gene expression data to the N-of-1 setting and show that this framework can identify distinct immune and stromal expression signatures that differentiate pediatric cancer samples. Finally, we identify recurrent tumor microenvironment signatures across pediatric cancer types associated with differences in patient outcomes.

## Materials and methods

### Dirichlet process gaussian mixture model

Traditional parametric models, like the finite mixture model, use a fixed number of parameters (i.e. number of clusters). Over- or underfitting can occur when the parametric model does not reflect the underlying data [48]. Unlike the finite mixture model, the Dirichlet process mixture model (DPMM) represents a theoretically infinite number of clusters and can adapt the number of clusters based on prior belief and the data [26, 48, 49].

The Dirichlet process (*DP*) is an infinite dimensional extension of the Dirichlet distribution [50] and is commonly used as a prior distribution for infinite mixture models [51, 52]. The Dirichlet process has two parameters: the concentration parameter $\alpha$ and centering distribution *H*. The concentration parameter $\alpha$, where $\alpha \in \mathbb{R}^+$, controls the extent to which samples from the *DP* resemble the centering distribution *H*. We model gene expression as a multivariate Gaussian distribution, so our centering distribution is a normal-Wishart distribution ($\mathcal{NW}_0$).

We briefly describe the stick-breaking construction of the Dirichlet process $G \sim DP(\alpha, H)$. Consider a stick of unit length. To generate an infinite number of mixing weights $\pi_1, \pi_2, \ldots, \pi_k$ for the DPMM, first break a stick of unit length at $v \in [0, 1]$ where $v$ is sampled from a Beta distribution, and set $\pi_1$ to be the length of the first piece. We repeat this process using the remainder of the stick for each $\pi_k$. The DP is truncated to the number of clusters K, which was shown to accurately approximate the infinite posterior for large K [26, 48, 50, 53–55].

$$v \sim \text{Beta}(1, \alpha) \tag{1}$$

$$\pi_k = v_k \prod_{l=1}^{k-1}(1 - v_l) \tag{2}$$

Next, we sample the parameters from the centering distribution *H* weighted by the mixing components. If we consider a probability space $\Theta$ where $\theta_k^* \in \Theta$, then *H* is a measure on the partitions of $\Theta$. For our application, we will partition the parameter space $\Theta$ into finite, measurable partitions $B_1, B_2, \ldots, B_k$.

$$\theta_k^* \sim H \tag{3}$$

$$G = \sum_{k=1}^{\infty} \pi_k \delta_{\theta_k^*} \tag{4}$$

$$(G(B_1), G(B_2), \ldots, G(B_k)) \sim \text{Dir}(\alpha H(B_1), \alpha H(B_2), \ldots, \alpha H(B_k)) \tag{5}$$

This construction generates the marginal of the Dirichlet process, which follows a Dirichlet distribution. Samples from the marginal distribution are finite, discrete, and sum to 1 [50]. Next, we outline how the DPMM groups gene expression samples $x_i$ under cluster-specific

parameters $\mu_{z_i}$ and $\Sigma_{z_i}$ where $z_i \in 1, 2, \ldots, K$ is the cluster index.

$$x_i | \mu_{z_i}, \Sigma_{z_i} \sim \mathcal{N}(\mu_{z_i}, \Sigma_{z_i}) \tag{6}$$

$$z_i | \pi \sim \text{Categorical}(\pi_1, \pi_2, \ldots, \pi_k) \tag{7}$$

$$\mu_{z_i}, \Sigma_{z_i} | G \sim G \tag{8}$$

$$G | \alpha, \mathcal{NW}_0 \sim DP(\alpha, \mathcal{NW}_0) \tag{9}$$

To improve our methods ability to scale to larger datasets, we incorporated the bnpy memoized online variational inference algorithm (moVB) [53] into our analysis framework. The moVB algorithm uses variational inference to approximate the posterior distribution and interleaves birth, merge, and delete moves to avoid local optima and remove redundant clusters [56]. We found that the moVB algorithm accurately identified the number of clustering on validation datasets (S1 Fig), whereas standard MCMC sampling procedures tended to overestimate the number of clusters.

## Hydra method

We developed a Bayesian non-parametric clustering framework for identifying biological and technical variation in large cancer gene expression datasets without the need for a reference normal dataset. To our knowledge, this is the first reproducible and widely deployable implementation of a non-parametric mixture model framework designed to overcome the challenges of precision oncology gene expression analysis. The hydra pipeline is an open source software tool hosted on GitHub (www.github.com/jpfeil/hydra). A Docker container is available for deployment across environments (https://hub.docker.com/r/jpfeil/hydra).

The hydra framework contains three main command-line tools: *filter*, *enrich*, and *sweep* (Fig 1). The *filter* command is run first to isolate the multimodally expressed genes using a univariate Dirichlet Process Gaussian Mixture Model (DP-GMM). There are two methods for analyzing the resulting set of multimodally expressed genes. The *enrich* method, which subsets to the genes found to be significantly enriched in biological pathways, and the *sweep* method, which searches within user-defined gene sets for multimodal expression signatures. The underlying analysis routines can be accessed within the Docker using Jupyter notebooks to facilitate the development of user-defined workflows.

The *filter* command (Fig 1B) takes an expression matrix and filters the genes down to the multimodally expressed genes using the DP-GMM described above. We apply a DP-GMM to each gene, saving the model for genes with two or more expression clusters. This creates a directory of multimodally expressed gene models which can be used to predict differential expression in new samples. This analysis framework is a novel contribution to the precision medicine research community. Our approach has several beneficial properties. For example, training models on curated data sets and applying the models to new samples avoids the use of reference distributions, which overestimate the uncertainty in the analysis by not accounting for subtype expression. Furthermore, this approach identifies the set of most strongly differentially expressed genes within a disease context, which may enrich for potential biomarkers for precision medicine applications. The multimodally expressed genes are also used in downstream clustering analysis.

The *enrich* (Fig 1C) and *sweep* (Fig 1D) routines are two independent analyses to explore multimodal expression in cancer gene expression cohorts. In addition to identifying

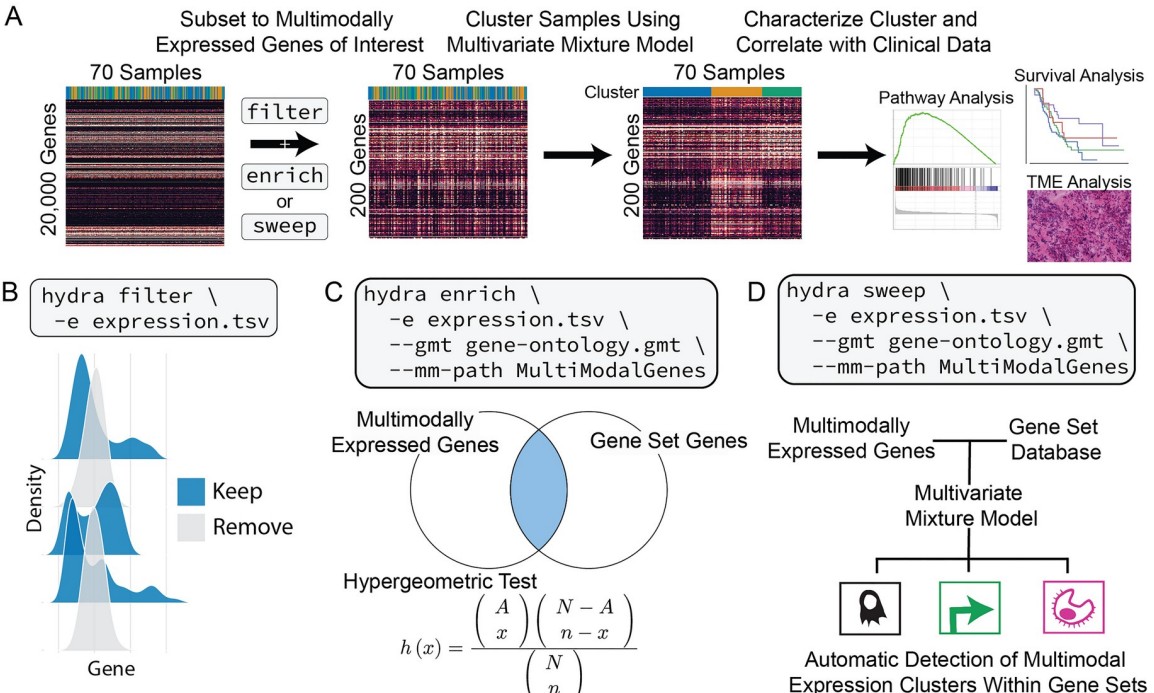

**Fig 1. Overview of the hydra framework tools.** A: Suggested workflow for applying hydra framework tools to identify clinically relevant gene expression subtypes. B: The hydra *filter* command removes unimodally distributed genes which greatly reduces the number of genes in downstream clustering analysis. C: The hydra *enrich* command takes the multimodally expressed genes and returns enriched gene sets. The enriched gene set genes are used for multivariate clustering of samples. D: The hydra *sweep* command looks for multivariate normal clusters within user-defined gene sets. This can be used for the automatic detection of clusters in large gene set databases. Abbreviations: Tumor microenvironment (TME).

expression variation within a disease context, we also found that multimodally expressed genes that participate in a biological pathway tend to have correlated expression distributions. This insight facilitates the detection of multimodal expression signatures by enriching for genes that have multimodal expression distributions and participate in known biological processes. The hydra software comes prepackaged with popular gene sets, including the Molecular Signatures Database (MSigDB) [18], the Gene Ontology terms [57, 58], and the EnrichmentMap gene sets [59]. The gene set database is configurable, so additional gene sets can be added at runtime.

The *enrich* command uses a hypergeometric test [60] to discover enrichment of multimodally expressed genes within a user-defined database of gene sets. This creates a list of gene sets and a list of enriched gene set genes. The *enrich* method outputs a table of enriched gene sets while also clustering samples across the genes that participate in the enriched gene sets. The table of enriched gene sets may reveal surprising expression patterns and generate hypotheses for further investigation of tumor subtypes.

The implementation of the *enrich* method includes an important parameter known as the minimum component probability. The minimum component probability is the probability of placing a sample within the smallest expression cluster. This is an additional filter to remove multimodally expressed genes that influence a relatively small subset of tumor samples. This parameter gives the user the ability to subset the enriched genes to those that influence a

greater number of patients. To aid in the exploration of minimum component thresholds, we implemented a *scan* sub-routine. The *scan* routine tunes the analysis with respect to the constraints of the available data (e.g. number of samples and number of genes), which is an important factor in pediatric cancer research since data is often difficult to obtain and so datasets are relatively small. We recommend setting this threshold such that the number of genes is less than the number of samples because otherwise the inference may become unstable [61].

The *sweep* routine identifies differentially expressed gene sets and can be used as an alternative to single-sample GSEA [16]. For each gene set, a multivariate DP-GMM is applied to determine if more than one expression cluster is present within the gene set. This approach is useful when curated gene sets are available for the disease of interest, but manual inspection of each gene set is not feasible. Reducing the genes to multimodally expressed genes facilitates the detection of differentially expressed gene sets. Existing gene set enrichment tools are known to under-perform when the expression is correlated [62], but our approach is designed to identify distinct correlation structures within gene expression datasets.

We have also implemented routines for cluster profiling and N-of-1 tumor analysis. These routines are accessible within the docker container using the Jupyter notebook command. Cluster profiling analysis of clusters derived from the *enrich* or *sweep* routines includes GSEA [63] to identify the pathway expression that characterizes each cluster. GSEA uses all available genes since it requires non-differentially expressed genes to assess the significance of an enrichment score. A t-statistic is calculated for each gene, comparing gene expression values of samples inside to those outside of a cluster. Cluster profiling GSEA uses the ranked gene-level t-statistics to determine gene set enrichment.

The N-of-1 tumor analysis routine classifies a new gene expression profile into one of the inferred clusters, calculates a gene-level z-score for that sample relative to the normalized expression distribution, and performs standard GSEA using a preranked list of z-score values [63]. This procedure can identify new gene expression signatures that may not be detectable using the entire expression cohort as a background reference distribution. This approach is another novel contribution to the field and may facilitate the identification of clinically relevant signatures that are being overlooked in current gene expression analyses.

## Synthetic data generation and validation

We first tested the hydra framework's ability to detect differential pathway expression using synthetic cancer data. We compared hydra *sweep* to two widely used gene set enrichment tools: single-sample gene set enrichment analysis (ssGSEA) and gene set variation analysis (GSVA) [64–66]. Both methods are implemented in the GSVA R package [65]. In order to accurately model correlation structures within cancer cohorts, we modeled the synthetic cancer gene expression data as a multivariate Gaussian distribution. We used the TCGA glioblastoma multiforme (GBM) cohort (N = 166) to model a background mean and covariance matrix for the synthetic data analysis. We chose TCGA GBM, a very different disease from those analyzed in the remainder of this manuscript, to avoid overfitting the hydra method to diseases of interest. This also enables us to demonstrate the flexibility of our method to analyze data from a variety of cancer genome sequencing projects.

This approach allowed us to model cancer gene expression data while also controlling for subtype-related expression variation. We downloaded the RSEM-quantified transcripts per million (TPM) normalized gene expression measurements from the UCSC Xena Browser [3]. We focus our analysis on normalized gene expression data because this data is more widely used in the cancer research community and fewer methods are available to analyze normalized

counts. To reduce heteroscedasticity and the effect of outlier expression levels, we transformed the expression data to log2(TPM + 1) [67].

We defined an expression subtype as a subset of samples with a distinct expression mean and correlation structure compared to other samples within the disease cohort. To avoid biases in the synthetic data generation process, we used random sampling to select MSigDB gene sets for each subtype, the size of the subtype, and the correlation structure within the subtype. We randomly generated a covariance matrix for the cancer subtype expression data, but used the underlying covariance matrix of the TCGA glioblastoma multiforme dataset for the background samples. We tested the effect of having 10% and 25% of genes within a gene set being differentially expressed (%DEG). In addition to these parameters, we tested a range of effect sizes: 0.25 (least different), 0.5, 0.75, 1.0, 1.5, 2.0, 2.5, and 3.0 (most different). This process was repeated twice for each gene set to create synthetic training and test data, which resulted in the generation of 640 synthetic datasets.

We then applied the hydra framework using the hydra *sweep* command (Fig 1C), since this method is directly comparable to the single-sample GSEA methods. The mean expression filter removed any genes with a mean expression of fewer than 1.0 log2(TPM + 1). This avoids lowly-expressed genes that may have particularly noisy expression measurements. The prior on the hydra covariance matrix was the identity scaled by 2.0 and the prior on the number of clusters was set to 2 because we expect there to be an activated cluster and a baseline expression cluster. We set the over-expressing cluster to be the cluster with the largest L1 norm.

## Pediatric cancer gene expression data

We downloaded pediatric cancer RNA-Seq data for neuroblastoma, osteosarcoma, Ewing sarcoma, alveolar rhabdomyosarcoma, and embryonal rhabdomyosarcoma from the UCSC Treehouse Compendium (https://treehousegenomics.soe.ucsc.edu/public-data/). This data was produced using the same RNA-seq pipeline, so potential computational batch effects are minimized [1, 6]. Clinical data for the TARGET neuroblastoma and osteosarcoma samples were obtained from the TARGET Data Matrix (https://ocg.cancer.gov/programs/target/data-matrix). We also analyzed a set of 58 synovial sarcoma microarray profiles with matching metastasis rate data [68].

## TARGET neuroblastoma analysis

We applied each hydra tool to the TARGET *MYCN*-non-amplified neuroblastoma cohort. We first obtained the multimodal gene models using the hydra *filter* tool. The hydra *filter* tool identified all genes with a multimodal expression pattern. We used the mean expression filter to remove genes that may have unstable measurements due to low transcript abundances. We excluded all genes with a mean expression value less than 1 log2(TPM + 1).

The hydra *sweep* command was applied to search for subtype expression within curated MSigDB gene sets. We included the hallmark (n = 50), BioCarta (n = 289), KEGG (n = 186), PID (n = 196), and Reactome (n = 1499) genesets [18]. We include all signatures with a minimum component probability of 10%. For example, the smallest subtype cluster considered in this analysis had 7 samples, since the total number of samples was 70. We investigated relationships among differentially expressed gene sets by clustering the gene sets by their pairwise Jaccard index. This created a similarity network that was then visualized using the Gephi software tool [69].

The hydra *enrich* command identified correlated expression signatures using the enriched GO term genes (FDR < 0.01). The multivariate mixture model $\alpha$ concentration parameter was set to 5.0; the prior on the covariance matrix was set to the identity scaled by 2.0. The

prior parameter for the number of clusters was set to 5. Our synthetic data analysis found that the signal decreases below an effect size of 1.0, so we use this parameter value for all following analyses. We used the hydra *scan* routine to search a range of minimum component probability thresholds (see Results) and found that a threshold/probability of 20% yielded the most clusters while keeping the number of genes (p = 42) below the number of samples (n = 70).

To validate tumor microenvironment expression subtypes, we correlated the hydra *enrich* expression clusters with the results of tumor microenvironment profiling tools xCell [31], CIBERSORT [70], and ESTIMATE [32]. We also compared the hydra *enrich* approach to state-of-the-art consensus clustering methods M3C [20] and k-means clustering using the Gap statistic to select the number of clusters [36]. Since these methods are influenced by the number of input genes, we tested a range of median absolute deviation (MAD) thresholds. The number of clusters was assumed to be the smallest statistically significant value.

### Small blue round cell tumor analysis

We then compared the clustering patterns across *MYCN*-NA neuroblastoma, osteosarcoma, Ewing sarcoma, embryonal rhabdomyosarcoma, alveolar rhabdomyosarcoma, and synovial sarcoma. We applied the TumorMap dimensionality reduction method [5] to visualize clustering of the full small blue round cell tumor gene expression matrix. We then applied the hydra framework to explore expression variation within each disease. Each disease expression matrix had unique statistical properties including sample size and subtype variation. This required us to adapt the minimum probability threshold for each disease dataset using the *scan* routine. The Jupyter notebooks for exploring these datasets can be found on GitHub (www.github.com/jpfeil/hydra-paper/analysis). We used agglomerative clustering to investigate patterns in the top 10 enriched gene sets for each disease's expression subtypes.

### Statistical analysis

A Kruskal-Wallis test was used to identify statistically significant differences across two or more groups, and a Mann-Whitney U test was used for pairwise tests using a Holm-Sidak correction for multiple hypothesis testing [71, 72]. We used the scipy [73] stats implementation of the Kruskal-Wallis test and the scikit-learn post hoc processing [74] implementation of pairwise Mann-Whitney U tests. Spearman rank and Pearson correlation values were calculated using the scipy library [72]. Survival analysis was done using the survminer package [75].

### H&E slide preparation and pathologist review

Pediatric tumor samples were flash frozen, embedded in OCT, and 5$\mu$m cryosections were collected. Slides were hematoxylin and eosin (H&E) stained and imaged on a Leica DMi8, equipped with a HC PL APO 40x/0.85 NA objective and DFC7000T camera. H&E slides were reviewed by a licensed pathologist. Morphologic analysis was performed and the degree and type of inflammation estimated from the histologic sections. Grading of inflammation was either minimal (<10% of total nuclei consist of inflammatory cells) or moderate (20-30% of total nuclei consist of inflammatory cells). The type of inflammation (predominantly small mature lymphocytes or mixed inflammation consisting of small mature lymphocytes along with plasma cells and/or eosinophils) was noted for each tumor sample.

## Results

### Performance assessment using synthetic gene expression data

To assess how well hydra detects differentially expressed pathways as compared to common pathway enrichment approaches, we applied the hydra framework to synthetically-generated cancer gene expression data. We generated synthetic cancer gene expression data based on the TCGA glioblastoma multiforme and the MSigDB Hallmark gene sets as described above. We tested a range of effect sizes and percent differentially expressed genes (%DEG) within the MSigDB gene sets. We generated receiver operator curves (ROC) and calculated the area under the receiver operator curve (AUC) for each analysis. Overall, the hydra pipeline outperformed the single-sample GSEA approaches with a mean AUC of 0.93 (95% CI: 0.91—0.95). ssGSEA had a mean AUC of 0.72 (95% CI: 0.71—0.74) and GSVA had a mean AUC of 0.67 (95% CI: 0.66—0.68) (Fig 2A).

We further investigated the performance of these methods by plotting the AUC against the effect size at 10 and 25%DEG (Fig 2B). The hydra method performed better across all effect sizes, achieving near perfect performance above an effect size of 2.0 and 0.75 at 10 and 25% DEG, respectively. ssGSEA and GSVA performed similarly at low effect sizes, but ssGSEA performed better than GSVA as the effect size increased. Overall, the hydra framework performed significantly better than these standard gene set enrichment approaches, particularly at low effect sizes. Therefore, the hydra approach is better suited for subtyping within a disease cohort when the effect sizes are smaller and fewer genes are differentially expressed.

We performed a runtime analysis comparing hydra *sweep*, ssGSEA, and GSVA for identifying a single differentially expressed gene set, since these methods are directly comparable. Training the hydra model was the most computationally expensive step, but the classification of new samples was very fast. The average runtime for the hydra *sweep* algorithm was similar to ssGSEA, but the hydra runtimes were more variable across effect-sizes and number of differentially expressed genes. The GSVA approach was faster than hydra *sweep* and ssGSEA, but GSVA performed worse on the synthetic data analysis than ssGSEA and hydra. We repeated the above analysis with an effect size of 1.0, a %DEG of 25%, and a range of sample sizes, including 50, 100, 200, 300, 400, 500, 1000 samples. The hydra *sweep* and GSVA methods scaled well to large sample sizes, but the ssGSEA runtime increased exponentially as the sample size increased (Fig 2C & 2D).

### Hydra analysis of high-risk neuroblastoma

High-risk neuroblastoma is an aggressive disease and is resistant to intensive therapy. Further subtyping of high-risk neuroblastoma may identify novel therapeutic targets and improve risk stratification. We hypothesized that unsupervised clustering of multimodally expressed genes associated with enriched Gene Ontology terms would identify expression subtypes of high-risk neuroblastoma tumors. TumorMap analysis [5] showed that the *MYCN*-non-amplified (*MYCN*-NA) neuroblastoma samples clustered separately from *MYCN*-amplified (*MYCN*-A) and stage 4S neuroblastoma samples (S2 Fig). We focused on the *MYCN*-NA neuroblastoma tumor samples because this is the largest set of samples (N = 70) and variation within *MYCN*-NA tumors is not well understood [28].

We applied the hydra *filter* analysis to the TARGET high-risk neuroblastoma cohort as described above. This analysis identified 931 genes within the *MYCN*-NA neuroblastoma cohort with a multimodal expression distribution. Of the 931 multimodally expressed genes, 358 genes were found to be potentially druggable by the Drug Gene Interaction Database (S1 File) and 60 genes were associated with an FDA-approved, anti-neoplastic drug [29].

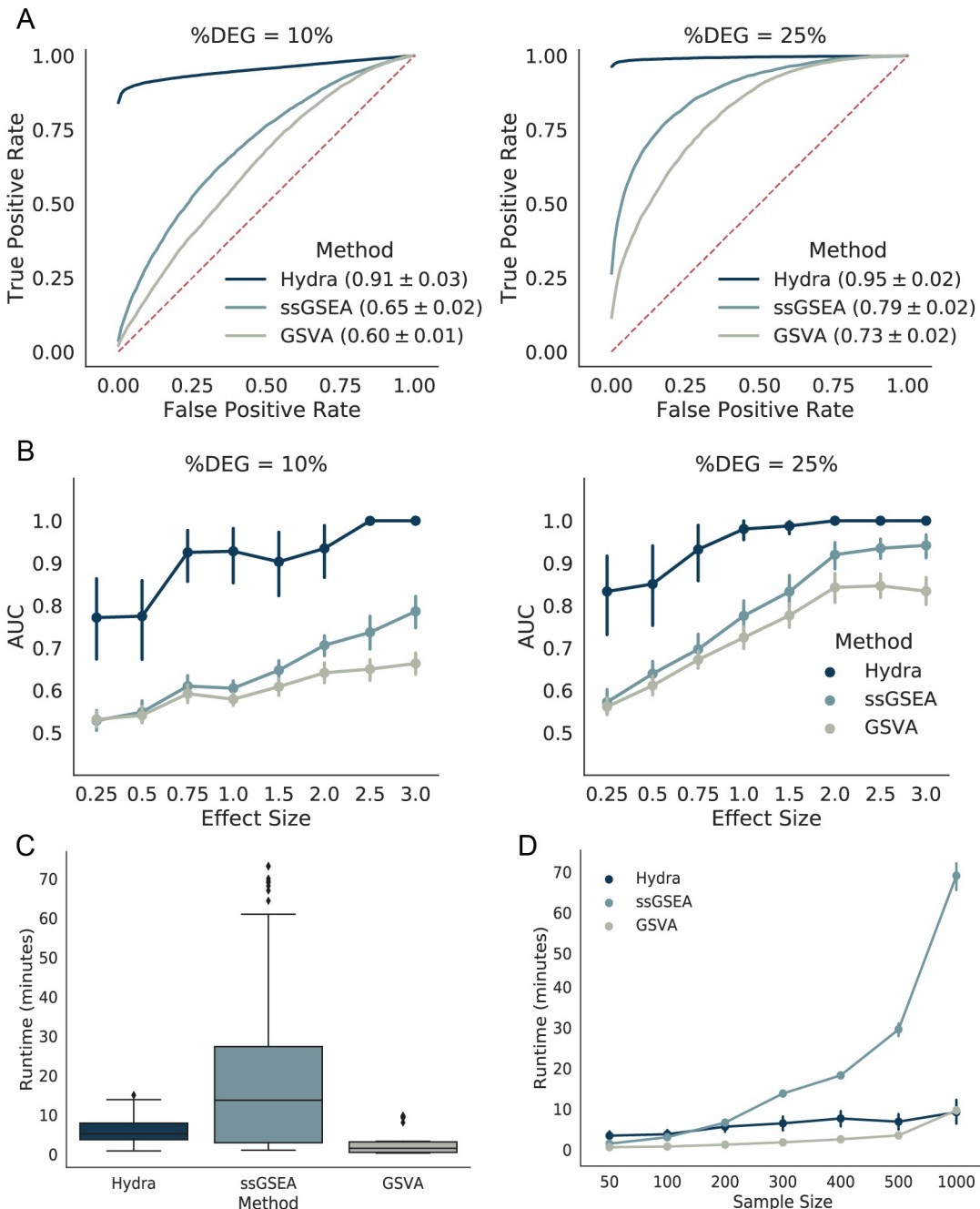

**Fig 2. Hydra** *sweep* **is more sensitive than existing gene set enrichment approaches for detecting differential pathway expression in synthetic data and scales well to large datasets.** A: Mean receiver operator curves across effect sizes, percent differentially expressed genes (%DEG), and MSigDB Hallmark gene sets. A larger area under the curve (AUC) indicates better performance. The average AUC and 95% confidence interval for each method are in the ROC plot figure legends. B: Line plots comparing the mean AUC across a range of effect sizes and %DEG values. C: Box plot showing mean runtimes for differential pathway analysis where the effect size is fixed but the sample size varies. D: Line plot comparing the mean runtimes for differential pathway analysis across a range of sample sizes.

We next examined whether unsupervised clustering of multimodally expressed genes revealed coordinated expression of annotated gene sets within the MSigDB database. Applying the hydra *sweep* command to the *MYCN*-NA neuroblastoma cohort discovered 105 gene sets with multimodal expression patterns. Each gene set sheds light on biological themes that are differentially expressed within the *MYCN*-NA neuroblastoma cohort. We clustered the differentially expressed gene sets to reveal these biological themes (S4 Fig). We found 6 major themes, including annotated cancer functions, cell cycle regulation, cell signaling pathways, immune functions, extracellular matrix reorganization, and metabolic pathway gene sets.

We applied the hydra *enrich* analysis to the MYCN-NA cohort to identify how the most highly enriched gene sets interact to form expression subtypes. This analysis found 428 genes with a minor component probability greater than 20% (S1 File). Gene Ontology analysis found enrichment for the following GO terms (FDR: q < 0.01): adaptive immune response (24 genes), mesenchyme development (12 genes), steroid hormone secretion (4 genes), and response to corticosterone (4 genes). DP-GMM analysis of the 44 enriched GO term genes identified three *MYCN*-NA neuroblastoma clusters (Fig 3A). The posterior probability for belonging to each cluster was 42%, 34%, and 17% for clusters 1, 2, and 3, respectively. The posterior probability for a sample belonging to a new cluster was about 6% in our analysis.

We next investigated cluster-specific expression signatures using GSEA (see Hydra Method section). Cluster 1 was enriched for adaptive immune response gene sets, cluster 2 was enriched for proliferative signaling gene sets, and cluster 3 was enriched for cancer-associated fibroblast gene sets (Fig 3B). Cluster 3 shares several features of a wound healing response, including fibroblast recruitment, extracellular matrix organization, and infiltration of immune cells [30].

Clusters 1 and 3 were enriched for tumor microenvironment-associated gene expression. To further validate this signal, we correlated the hydra clusters with enrichment scores from the tumor microenvironment profiling tools xCell [31] and ESTIMATE [32]. Cluster 1 had high average xCell enrichment scores associated with adaptive immune cell types including B-cells, CD4+ naive T-cells, and CD8+ naive T-cells (Kruskal-Wallis: p < 0.001). Cluster 2 was characterized by the absence of immune and stromal expression and higher tumor purity scores than clusters 1 and 3. The average ESTIMATE tumor purity was 88%, 96% and 82% for clusters 1, 2, and 3, respectively. Cluster 3 was enriched for fibroblast-associated expression by xCell analysis (Kruskal-Wallis: p < 0.001). Clusters 1 and 3 had higher ESTIMATE immune-associated expression levels than cluster 2 (average ImmuneScore per cluster: 58, -612, 56), but cluster 3 had the highest stromal expression signature score (average StromalScore per cluster: -1027, -1310, -135). Comparing ESTIMATE enrichment scores across clusters reveals clear trends in broad immune and stromal expression signatures. Lastly, we found a correlation between the hydra-identified tumor microenvironment subtype and *CD274* and *CTLA4* expression (S6 Fig).

We next correlated clusters with clinical features. We found no difference in patient survival outcomes across clusters (log-rank test, p > 0.05). Notably, cluster 1, which had the highest adaptive immune expression signal in MYCN-NA neuroblastoma, over-expresses cell-cycle regulation genes, which was not observed in other small blue cell tumors. We investigated associations with clinical covariates, including mutation burden, age, and tumor content as assessed by a clinical pathologist, but found no statistically significant differences (Kruskal-Wallis: p > 0.05). We then investigated associations between the hydra clusters and

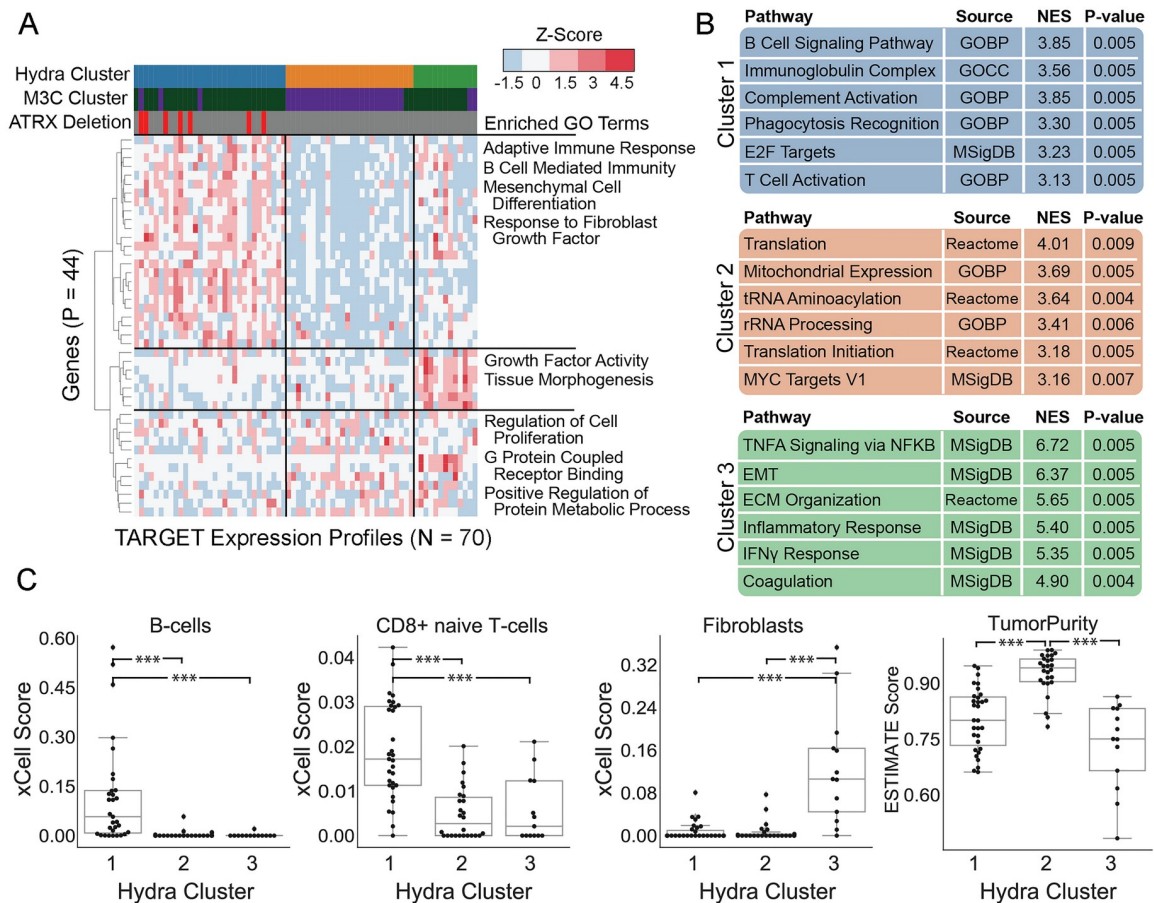

**Fig 3. Hydra analysis identifies three distinct tumor microenvironment expression subtypes in *MYCN* non-amplified neuroblastoma samples.** A: Gene expression heatmap displaying expression profiles of hydra clusters. Heatmap columns (samples) are ordered by hydra cluster membership. Ward hierarchical clustering applied to rows (genes) identified coordinated expression of GO term genes. These GO term genes were originally identified by the hydra *enrich* command. B: GSEA performed on each cluster identified enrichment of tumor microenvironment and proliferative signaling gene sets. C: xCell enrichment score distributions for B-cells, CD8+ naive T-cells, and Fibroblasts, and the ESTIMATE TumorPurity score distributions for each cluster; enrichments for all cell types are available in S1 File. Abbreviations: Normalized Enrichment Score (NES), Epithelial to Mesenchymal Transition (EMT), Extracellular Matrix (ECM), Gene Ontology Biological Process (GOBP).

neuroblastoma-associated molecular aberrations and clinical features (S1 File). *ATRX* gene deletions were enriched in cluster 1 (Fisher's Exact Test: p < 0.05). MKI low tumors were enriched in cluster 2 and 3 (Fisher's Exact Test: p < 0.01). Chromosome 17 wild-type tumors were enriched in clusters 2 and 3 (Fisher's Exact Test: p < 0.01). Analysis on a larger dataset may reveal additional clusters and correlations with clinical features.

Consensus clustering is a widely used approach for identifying tumor subtypes using gene expression data. We applied the M3C consensus clustering method, which is a more sophisticated version of consensus clustering that uses a null distribution to assess the statistical significance of the clustering [20, 21]. We used the top 5000 genes with the largest median absolute deviation (MAD) because this threshold is routinely used in unsupervised clustering of cancer gene expression data [33–35].

The M3C analysis resulted in the identification of two statistically significant clusters. One M3C cluster correlated with hydra clusters 1 and 3 and the other M3C cluster correlated with

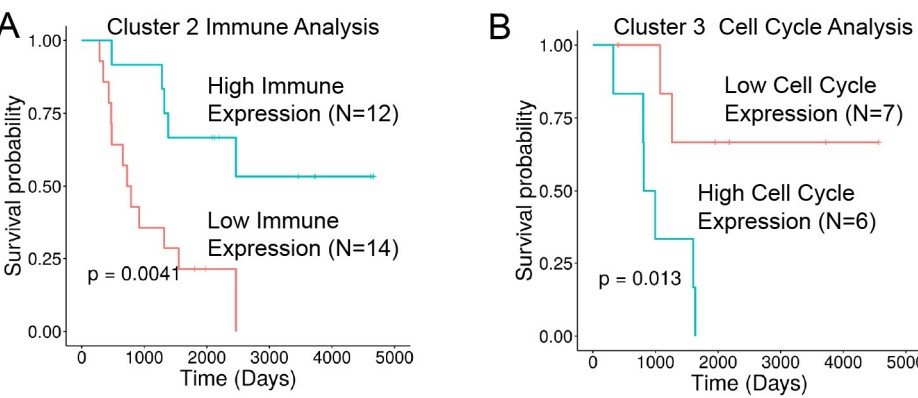

**Fig 4. Gene set enrichment analysis (GSEA) of *MYCN*-NA neuroblastoma identifies overall survival differences within hydra cluster 2 and cluster 3.** Cluster-level GSEA separated cluster 2 into high and low immune expression subtypes and cluster 3 into high and low cell cycle expression subtypes. A: Kaplan-Meier plot for immune expression subtypes within cluster 2. B: Kaplan-Meier plot comparing cell cycle expression subtypes within cluster 3.

hydra cluster 2. Therefore, M3C clustering detected the tumor purity signal in the expression data, but was not able to separate the adaptive immune cell and fibroblast infiltrated clusters (hydra clusters 1 and 3). We also applied k-means clustering using the gap statistic approach [36, 37] for estimating the number of clusters, but this approach grouped all samples into a single cluster. We tested a range of MAD thresholds based on the median absolute deviation, but found similar results across thresholds (S3 Fig). Overall, the hydra approach was more sensitive at detecting distinct tumor microenvironment states than these other popular clustering methods.

To further investigate expression patterns within the hydra-identified tumor microenvironment subtypes, we performed GSEA by z-score normalizing each tumor's gene expression data to its tumor microenvironment cluster. This is a novel GSEA approach that uses the tumor microenvironment state discovered by the hydra method to identify additional gene expression signals for individual samples. This approach revealed signals not present at the cohort level analysis (Fig 4). For example, enrichment of immune expression signatures within cluster 2 predicted differences in overall survival such that patients with higher immune expression had a better overall survival rate. Similarly, an elevated cell cycle signal within cluster 3 predicted worse survival compared to other cluster 3 samples with lower cell cycle expression. A metastatic expression signal was identified in the analysis of cluster 1 samples, but this signature did not correlate with a difference in survival. This approach may therefore provide appropriate background distributions for revealing and evaluating the significance of gene expression patterns and survival statistics within tumor subtypes.

## N-of-1 tumor analysis for pediatric neuroblastoma

The command-line interface of the hydra toolkit includes a *predict* function for labeling samples using a pre-fit model. The *MYCN*-NA neuroblastoma model described above was used to predict expression subtypes on a new set of samples. We obtained tumor gene expression data from six stage 4, *MYCN*-NA neuroblastoma samples from the UCSC Treehouse gene expression compendium [5, 6]. The age at diagnosis ranged from 2 to 6 years. Four out of six samples had a deletion in the *ATRX* gene.

Application of the hydra N-of-1 analysis framework clustered 4 out of the 6 samples into cluster 1, which is characterized by adaptive immune cell expression. Three of the *ATRX*-

deleted samples clustered with the high adaptive immune cell expression cluster (cluster 1) and one clustered in the low immune, high proliferative signaling cluster (cluster 2). We showed earlier that tumors with *ATRX* deletions tend to have higher adaptive immune expression, and we found a similar pattern in an independent set of *MYCN*-NA neuroblastoma samples.

Two of the samples with loss of *ATRX* came from the same patient but at different timepoints. The first sample (diagnostic sample) clustered with high adaptive immune cell expression (cluster 1), but the resection sample clustered with the low immune expression, high proliferative signaling cluster (cluster 2). We investigated possible explanations for the change in tumor microenvironment state. We performed GSEA comparing the samples from different timepoints to investigate potential mechanisms leading to immune evasion in these samples. GSEA found downregulation of the MHC Class I Antigen Processing & Presentation GO term in the resection sample (adjusted p-value < 0.002). Loss of antigen processing functions is a common mechanism of immune evasion across cancer types [38].

We obtained H&E stained sections for each of the hydra-identified clusters (S5 Fig). The cluster 1 sample had moderate levels of inflammation (30-50%) consisting of mature mononuclear cells, plasma cells, and eosinophils. The cluster 2 sample had minimal levels of inflammation (<10%) with some scattered mature mononuclear cells throughout the tumor. The cluster 3 sample looked similar to the cluster 1 slide with moderate levels of inflammation (30-50%), but also had regions of apparent necrosis. The inflammation and necrosis in the cluster 3 sample may correlate with the tissue remodeling/wound healing signature identified in the expression data.

## Hydra analysis discovers complex tissue signatures

While the *MYCN*-NA neuroblastoma analysis above focused on immune and wound healing expression signatures, the hydra *enrich* method is unsupervised and can therefore detect any type of expression signature. To illustrate this, we applied the hydra *filter/enrich* analysis to the TARGET osteosarcoma cohort (N = 74) and discovered enrichment of the GO striated muscle contraction term (FDR < 0.01, Fig 5). Multivariate clustering for the GO striated muscle contraction gene set using the *sweep* routine identified two clusters. xCell analysis of the osteosarcoma cohort found significant enrichment of skeletal muscle expression in the second cluster (Mann-Whitney U test, p < 0.001). Surprisingly, the M3C clustering approach was not able to detect the strong muscle signature using the 5000 genes with the largest MAD (p > 0.05). We used the muscle expression signature to identify osteosarcoma tumors in the UCSC Treehouse Compendium which also contained a similar expression signature. We subsequently confirmed with a licensed pathologist that one of the muscle-expression positive tumor samples did contain significant muscle tissue infiltration. The hydra *enrich* analysis revealed expression signatures not routinely investigated when analyzing osteosarcoma data. Nevertheless, these signals contribute significantly to the tumor expression profile, so explaining these sources of variation is necessary to derive clinically relevant conclusions from gene expression data.

We applied the *filter* method to Ewing sarcoma and discovered multimodal expression of an important druggable gene, JAK1. Applying the multimodal expression model allowed us to deconstruct the Ewing sarcoma distribution into three components (S7 Fig). We found that the expression component with the highest JAK1 expression was also enriched for mast cell expression. Therefore, overexpression of JAK1 may not correspond to activation of the JAK/STAT signaling pathway in cancer cells but rather to the presence of mast cells within the tumor microenvironment. Furthermore, targeted inhibition of JAK1 using ruxolitinib was shown to inhibit essential mast cell functions, including degranulation [39]. Therefore,

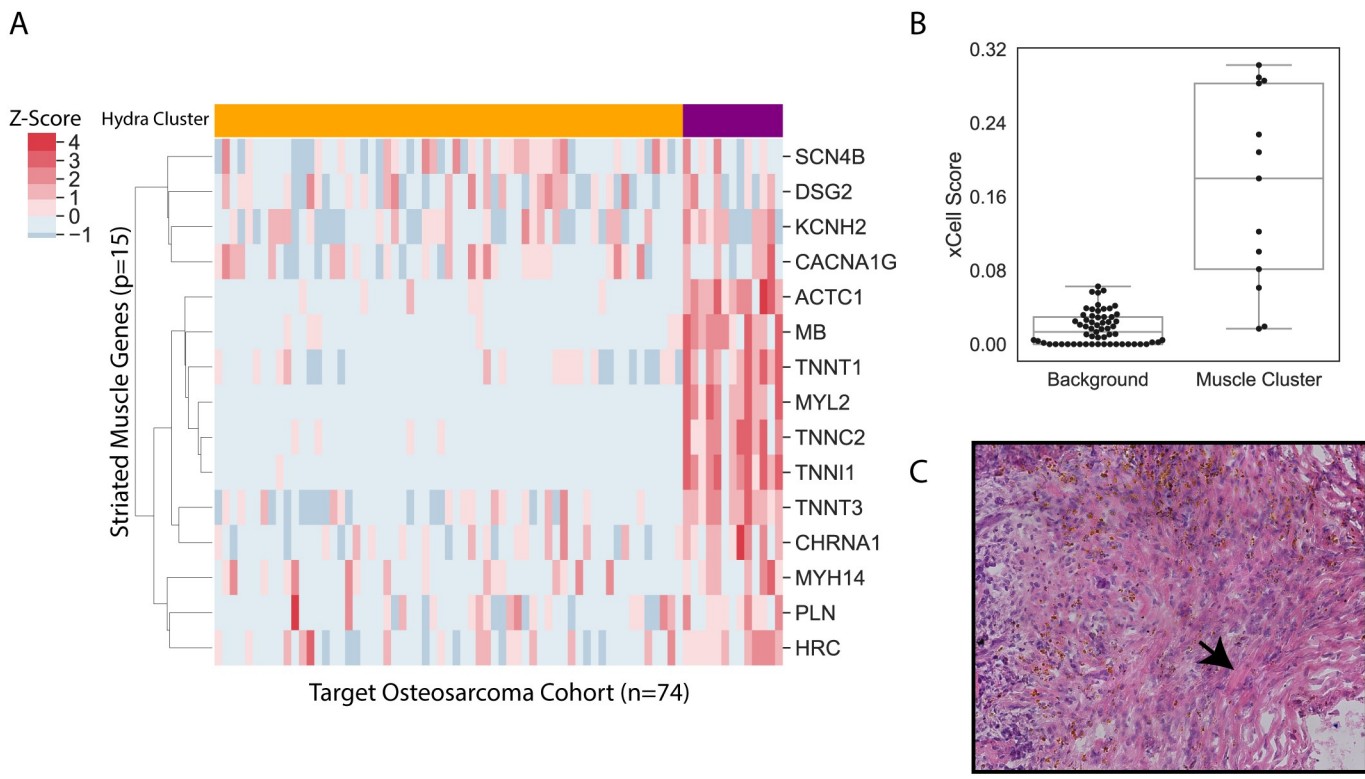

**Fig 5. Hydra analysis of TARGET osteosarcoma cohort reveals skeletal muscle signature.** Hydra enrichment analysis on the TARGET osteosarcoma cohort revealed a subset of patients with high skeletal muscle expression. A: Clustered heatmap shows the muscle signature genes identified by hydra unsupervised enrichment analysis (purple: enriched for muscle signature; yellow: not enriched for muscle signature). B: xCell tumor microenvironment profiling identified significant differences in skeletal muscle expression compared to background (p < 0.001). C: H&E stained tumor slide confirms presence of striated muscle tissue within the tumor sample.

therapeutic intervention intending to inhibit JAK1 expression in cancer cells may inadvertently inhibit the patient's mast cell functions. Overexpression analysis using the Ewing sarcoma JAK1 expression distribution may identify JAK1 as an actionable lead, but further investigation into the effect of inhibiting off-target JAK1 expression in mast cells is needed. The hydra framework facilitates the identification of important expression signatures which can be used to deconstruct complex tumor expression subtypes and identify potentially confounding expression signals.

We next quantified the number of multimodal druggable genes from the *MYCN*-NA neuroblastoma dataset that correlated with at least one xCell cell type signature. Out of the 358 druggable genes, we found that 77 correlated with a non-cancer cell type (Kruskal-Wallis test: Holm-Sidak adjusted p-value < 0.05, S1 File). Some of the druggable genes were expected to correlate with non-cancer cells, including the cytokines *IL6* and TGFB2, which correlated with epithelial cells and fibroblasts, respectively. Other druggable genes were surprising, like *AURKA* and *AURKB*, which correlated with higher Th2 cell expression. Aurora kinases play essential roles in spindle formation during mitosis and the overexpression of these genes is associated with evading spindle formation checkpoints in cancer [40], but little is known in how these genes correlate with infiltrating immune cells. Aurora kinase inhibitors show limited clinical activity in solid tumors, but have been shown to have a greater effect in leukemias [40, 41].

## Hydra analysis reveals recurrent expression subtypes across small blue round cell tumors

We next investigated whether similar hydra clusters could be identified across other small blue round cell tumors. We chose to focus on extracranial solid tumors because they are among the most common pediatric cancers, making up 20% of all pediatric cancer diagnoses [42], and while survival rates have improved, there are few effective treatment options for the subset of patients with relapse or refractory disease [43]. Identifying expression subtypes for these diseases may improve risk stratification and discover opportunities for new therapies. These tumors also share similar histopathological features, so we hypothesized that these tumors may share similar gene expression subtypes, despite significant differences in the raw expression profiles (Fig 6A).

We first performed TumorMap analysis, which is a dimensionality reduction approach for visualizing genomic data on a 2D surface [5]. We found that small blue round cell tumor types—*MYCN*-NA neuroblastoma, osteosarcoma, Ewing sarcoma, synovial sarcoma, alveolar rhabdomyosarcoma, and embryonal rhabdomyosarcoma—all form separate TumorMap clusters (Fig 6A). This suggests there is a strong cell-of-origin signal driving the clustering of these cancer types, which is an observation that was recently made in the larger TCGA dataset of adult cancers [44]. While pan-cancer analysis emphasized the differences across small blue round cell tumors, we hypothesized that expression subtypes within cancer types would participate in shared biological themes.

We next performed hydra *enrich* analysis within each small round blue cell cancer type and found shared biological themes across all six small blue round tumor types. Hierarchical clustering of the top 10 statistically significant gene sets for each cancer type resulted in clustering by expression subtype and not the cancer type (Fig 6B). Common themes emerged across

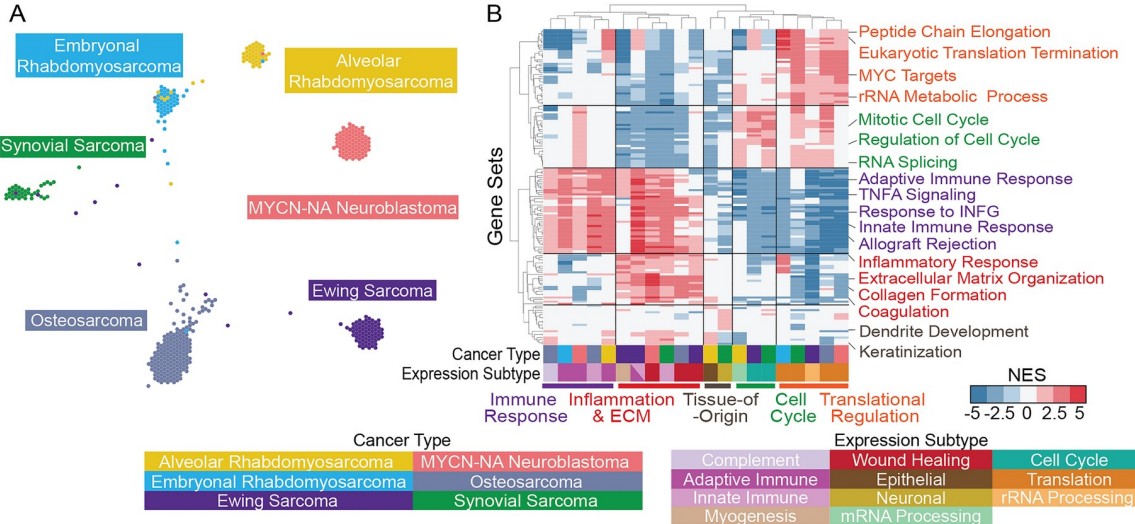

**Fig 6. Hydra *enrich* analysis of small blue round cell tumors reveals similar expression subtypes across cancer types.** A: TumorMap visualization of 6 small blue round cell tumor types. B: Hierarchically clustered heatmap for the top 10 enriched gene sets across the 21 small blue round cell tumor expression subtypes. Each column corresponds to a cancer type and an expression subtype (x-axis). Each row corresponds to a gene set. The expression subtype was manually assigned after reviewing the most highly enriched gene sets for each cancer expression subtype.

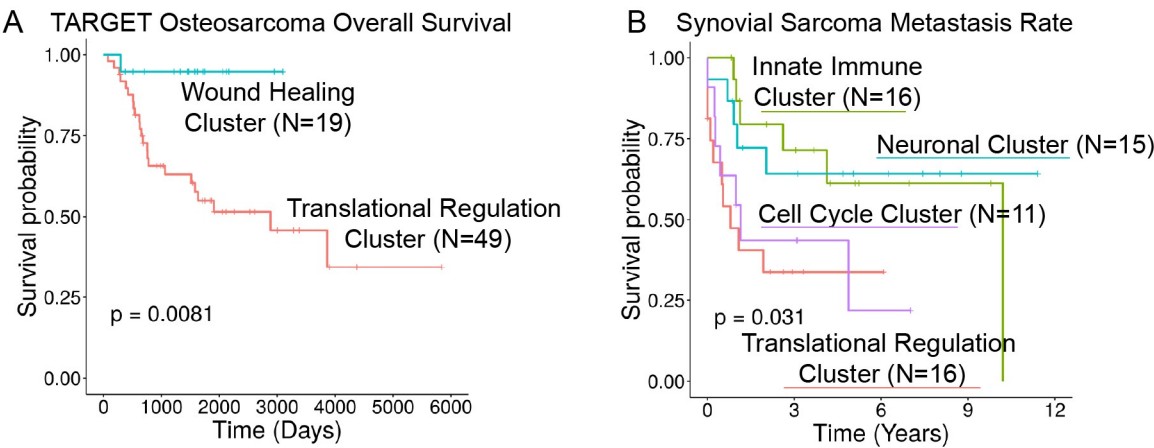

**Fig 7. Hydra analysis identifies tumor microenvironment expession subtypes that correlate with patient outcomes in osteosarcoma and synovial sarcoma.** A: Kaplan-Meier plot showing overall survival curves for osteosarcoma wound healing and translation clusters. B: Kaplan-Meier plot showing metastasis survival curves for synovial sarcoma clusters.

diseases including translational regulation, cell cycle regulation, immune effector cell signaling, inflammation, extracellular matrix organization, and tissue-of-origin signals. Furthermore, these signals predicted differences in patient outcomes in osteosarcoma and synovial sarcoma (Fig 7). In both cases, the presence of immune-associated expression correlated with better patient outcomes compared to tumors with proliferative signaling pathways associated with translation initiation and cell cycle regulation. Other osteosarcoma clusters were not included in the survival analysis due to insufficient number of samples with survival data (n < 5). Survival data were not available for the rhabdomyosarcoma and Ewing sarcoma expression datasets.

## Discussion

The hydra framework uses model-based clustering to facilitate the discovery of recurrent expression patterns within cancer gene expression cohorts. We leveraged recent improvements in model-based clustering algorithms to identify differentially expressed genes without a matched normal distribution. We modeled differential expression as a multimodal Gaussian distribution using nonparametric Bayesian statistics. We then enriched for biologically-annotated Gene Ontology terms and performed multivariate clustering to reveal expression subtypes. The hydra framework can be used for both identifying expression subtypes within large cohorts and classifying new tumor gene expression profiles using the trained models. The hydra framework outperformed standard gene set enrichment tools for identifying overexpression of the MSigDB Hallmark cancer gene sets in synthetic data. Application of this framework to small blue round cell tumors identified shared biological themes associated with the tumor microenvironment.

Multivariate gene expression analysis is typically underpowered because the number of genes greatly exceeds the number of samples. To address this limitation, we propose selecting for multimodally expressed genes before performing multivariate analysis. The hydra *filter* method reduces the number of genes and enriches for genes that participate in known biological processes, including those curated in the Gene Ontology and MSigDB databases. Selecting for multimodally expressed genes improves separation of known clinical subtypes better than the standard approach of using all expressed genes according to TumorMap analysis (S2 Fig). We also showed that the hydra approach of subsetting to multimodal genes improves detection

of differential pathway expression, including the identification of expression subtypes associated with the TME.

Significant progress has been made in subtyping neuroblastomas and adapting therapy for aggressive subtypes, but unexplained heterogeneity remains [28]. Failure to account for this heterogeneity decreases the power of standard methods to detect important expression patterns. Identifying biomarkers using genome-wide technology may lead to improved risk stratification and the discovery of novel drug targets. Hydra analysis of the TARGET *MYCN*-NA neuroblastoma cohort found differential expression of tumor microenvironment markers, including markers of the adaptive immune response. Pediatric cancers are generally thought to be less immunogenic because they have lower mutation burdens than adult cancers, but the immunogenicity of pediatric cancer has not been sufficiently investigated [11, 12].

Our analysis found significant variation in immune marker expression, including markers of response to checkpoint blockade therapy, and identified *ATRX* deletions as a potential biomarker of immune infiltrated tumors in *MYCN*-NA neuroblastoma. Analysis of other small blue round cell tumors revealed similar expression signatures across tumor types, despite samples clustering by their histology in a pan-cancer TumorMap analysis. Identification of shared expression signatures across cancer types may suggest that these patients would respond similarly to therapies that target these pathways. In particular, the identification of a cross-disease subtype associated with high expression of immune markers may warrant further investigation of immunotherapies in small blue round cell tumors using a basket clinical trial design [45].

Hydra analysis found significant differences in tumor immune and stromal expression that may inform precision medicine applications. The tumor microenvironment has become an important therapeutic consideration, but few methods account for the tumor microenvironment directly. Tumor purity has been identified as a confounding factor in cancer gene expression subtyping efforts [46]. For example, tumor purity and tumor microenvironment expression have been shown to correlate with pancreatic cancer subtypes [47]. Furthermore, Aran et al. (2018) found that tumor purity was correlated with the mesenchymal glioblastoma subtype and recommended a differential expression approach to computationally remove the tumor purity signal. However, standard approaches for subtracting the tumor purity effect may not be ideal because several mechanisms may influence tumor purity, and each mechanism may result in a different expression pattern. For instance, our analysis of *MYCN*-NA neuroblastoma identified two gene expression signatures that correlated with lower predicted tumor purity. Cluster 1 had an adaptive immune expression signature and cluster 3 had a cancer-associated fibroblast signature. Therefore, the estimated tumor purity signal should not be subtracted without first accounting for the different mechanisms influencing tumor purity.

We also found shared biological pathway enrichment across small blue round cell tumors. While these diseases are related and may derive from similar cell lineages, current expression methods often emphasize difference across these diseases (Fig 6A). Unsupervised clustering of adult cancer types found that cell-of-origin signals strongly influence clustering of cancer gene expression data [44]. Although these diseases have distinct expression patterns on the surface, we discovered common themes once we subset the data to the cell-of-origin signal and applied the hydra analysis tools.

We found at least three shared TME states: immune silent, immune infiltrated, and wound healing subtypes. The wound healing subtypes predicted better overall survival in osteosarcoma and delayed metastases in synovial sarcoma tumors, which suggests the involvement of the host immune response limits the progression of these tumors. Amplification of the host immune response may further limit tumor growth and lead to immune-mediated tumor cell death. Additional research into immune modulating therapies is warranted in small blue round cell tumors and may lead to improved outcomes for some patients.

## Conclusion

Precision oncology aims to differentiate tumors of the same diagnosis in order to match patients with the best treatment. We have developed the hydra framework to discover subtle but recurrent expression patterns within a cohort of samples with the same diagnosis, which is a novel strategy for pediatric precision oncology research. Our approach may help to uncover the biology underlying tumor progression and response to therapy. We have shown that hydra is more sensitive than standard gene set enrichment approaches for detecting differential pathway expression. Additionally, our framework provides tools to conduct unsupervised clustering analysis to discover expression subtypes. We applied the unsupervised hydra analysis to small blue round cell tumors and discovered distinct tumor microenvironment (TME) states. This shows that one of the strongest signals in clinical gene expression data comes from the TME, so careful modeling of the TME is required to maximize the impact of clinical gene expression analysis. The hydra framework provides unbiased clustering tools to characterize these sources of variation in specific disease populations and identify shared biological themes that can potentially be targeted therapeutically.

## Supporting information

**S1 Fig. Example of bnpy memoized online variational inference clustering on toy data.** We used the bnpy moVB algorithm to infer the number of clusters from synthetic data. The model first randomly assigns clusters. Then, the model iteratively improves the model fit, creating and destroying clusters until the model converges on the correct number of clusters at lap 16 [56].
(TIF)

**S2 Fig. Enriching for multimodally expressed genes improves clustering of established neuroblastoma subtypes.** Standard TumorMap analysis of the TARGET neuroblastoma dataset resulted in stage 4S samples clustering with stage 4 neuroblastoma samples (left). An alternative TumorMap based solely on 1,498 multimodally expressed genes separated the stage 4S samples into a distinct cluster (right).
(TIF)

**S3 Fig. Consensus and k-means clustering applied to TARGET *MYCN*-NA dataset.** We tested a range of gene expression variation thresholds based on the median absolute deviation, but found that the clusters identified by this approach could not resolve the same clusters as the hydra approach. The barplot shows the number of clusters and the lineplot tracks the Rand index comparing the M3C and k-means clusters and the hydra clusters.
(TIF)

**S4 Fig. Hydra *sweep* analysis reveals differential pathway expression within *MYCN*-NA neuroblastoma without a matched cohort of normal tissue.** Unsupervised clustering of multimodal gene sets revealed biological themes associated with hallmark cancer functions, including cell cycle, immune cell signaling, extracellular matrix organization, and metabolism.
(TIF)

**S5 Fig. Hydra method correlates with distinct tumor features as assessed by licensed pathologist review of tumor H&E slides.** A-B: H&E sections from fresh frozen tumor tissue from *MYCN*-NA neuroblastoma sample at A: 2X magnification and B: 20X magnification. Tumor cells are medium to large with moderate amounts of cytoplasm and areas of rhabdoid appearing undifferentiated cells. There is a moderate amount of mixed inflammation present (30-50%) consisting mostly of mature mononuclear cells with some plasma cells and scattered

eosinophils. C-D: H&E sections from fresh frozen tumor tissue from *MYCN*-NA neuroblastoma at C: 2X magnification and D: 20X magnification. Tumor cells are moderate to large in size with moderate amounts of cytoplasm. There is a minimal amount (<10%) of apparent mononuclear inflammation scattered throughout the tumor. E-F: H&E sections from fresh frozen tumor tissue from *MYCN*-NA neuroblastoma sample at (E) 2X magnification and (F) 20X magnification. Tumor cells are medium to large with moderate amounts of cytoplasm and areas of rhabdoid appearing undifferentiated cells. There are also areas of apparent necrosis. There is a moderate amount of inflammation present (30-50%) consisting mostly of mature mononuclear cells with some plasma cells and scattered eosinophils.
(TIF)

**S6 Fig. Hydra *enrich* analysis identifies correlation between expression subtypes and checkpoint blockade markers in *MYCN*-NA neuroblastoma.**
(TIF)

**S7 Fig. Hydra analysis identified *JAK1* expression clusters that correlate with mast cell expression signature in Ewing sarcoma.** A: *JAK1* expression distribution for Ewing sarcoma cohort (top) and the JAK1 expression distributions for cluster 1 (green), 2 (orange), and 3 (blue). B: Boxplot showing the xCell mast cell enrichment score for the three clusters associated with *JAK1* expression.
(TIF)

**S1 File. TARGET *MYCN*-NA neuroblastoma supplementary data.**
(XLSX)

**S2 File. Hydra method documentation.**
(PDF)

## Acknowledgments

We would like to thank the patients and families who participate in pediatric oncology research.

## Author Contributions

**Conceptualization:** Jacob Pfeil, Lauren M. Sanders, Ioannis Anastopoulos, A. Geoffrey Lyle, Alana S. Weinstein, Yuanqing Xue, Holly C. Beale, Alex Lee, Phuong T. Dinh, Avanthi Tayi Shah, Sofie R. Salama, E. Alejandro Sweet-Cordero, David Haussler, Olena Morozova Vaske.

**Data curation:** Jacob Pfeil, A. Geoffrey Lyle, Yuanqing Xue, Holly C. Beale, Stanley G. Leung, Phuong T. Dinh, Avanthi Tayi Shah, Marcus R. Breese, W. Patrick Devine, E. Alejandro Sweet-Cordero, Olena Morozova Vaske.

**Formal analysis:** Jacob Pfeil, Lauren M. Sanders, Ioannis Anastopoulos, A. Geoffrey Lyle, Yuanqing Xue, Andrew Blair, Holly C. Beale, Stanley G. Leung, Phuong T. Dinh, Marcus R. Breese, W. Patrick Devine, Olena Morozova Vaske.

**Funding acquisition:** Isabel Bjork, Sofie R. Salama, David Haussler, Olena Morozova Vaske.

**Investigation:** Jacob Pfeil, Lauren M. Sanders, Ioannis Anastopoulos, Yuanqing Xue, Andrew Blair, Holly C. Beale, Alex Lee, Stanley G. Leung, Phuong T. Dinh, Avanthi Tayi Shah, Marcus R. Breese, W. Patrick Devine, E. Alejandro Sweet-Cordero, David Haussler, Olena Morozova Vaske.

**Methodology:** Jacob Pfeil, Ioannis Anastopoulos, A. Geoffrey Lyle, Alana S. Weinstein, Yuanqing Xue, Andrew Blair, Holly C. Beale, Alex Lee, Sofie R. Salama, E. Alejandro Sweet-Cordero, David Haussler, Olena Morozova Vaske.

**Project administration:** Isabel Bjork, Sofie R. Salama, David Haussler, Olena Morozova Vaske.

**Resources:** Isabel Bjork, Sofie R. Salama, E. Alejandro Sweet-Cordero, David Haussler, Olena Morozova Vaske.

**Software:** Jacob Pfeil, Yuanqing Xue, Andrew Blair, Alex Lee, Isabel Bjork, David Haussler.

**Supervision:** Holly C. Beale, W. Patrick Devine, Isabel Bjork, Sofie R. Salama, E. Alejandro Sweet-Cordero, David Haussler, Olena Morozova Vaske.

**Validation:** Jacob Pfeil, Alex Lee, Stanley G. Leung, Phuong T. Dinh, Avanthi Tayi Shah, Marcus R. Breese, W. Patrick Devine, E. Alejandro Sweet-Cordero, Olena Morozova Vaske.

**Visualization:** Jacob Pfeil, Yuanqing Xue, Andrew Blair, Phuong T. Dinh, W. Patrick Devine, E. Alejandro Sweet-Cordero, Olena Morozova Vaske.

**Writing – original draft:** Jacob Pfeil.

**Writing – review & editing:** Jacob Pfeil, Lauren M. Sanders, Ioannis Anastopoulos, A. Geoffrey Lyle, Alana S. Weinstein, Yuanqing Xue, Andrew Blair, Holly C. Beale, Alex Lee, Stanley G. Leung, Phuong T. Dinh, Avanthi Tayi Shah, Marcus R. Breese, W. Patrick Devine, Isabel Bjork, Sofie R. Salama, E. Alejandro Sweet-Cordero, David Haussler, Olena Morozova Vaske.

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
