## [Decision Letter · Decision Letter 0]

11 Dec 2019

Dear Dr Pfeil,

Thank you very much for submitting your manuscript, 'Hydra: A mixture modeling framework for subtyping pediatric cancer cohorts using multimodal gene expression signatures', to PLOS Computational Biology. As with all papers submitted to the journal, yours was fully evaluated by the PLOS Computational Biology editorial team, and in this case, by independent peer reviewers. The reviewers appreciated the attention to an important topic but identified some aspects of the manuscript that should be improved.

We would therefore like to ask you to modify the manuscript according to the review recommendations before we can consider your manuscript for acceptance. Your revisions should address the specific points made by each reviewer and we encourage you to respond to particular issues Please note while forming your response, if your article is accepted, you may have the opportunity to make the peer review history publicly available. The record will include editor decision letters (with reviews) and your responses to reviewer comments. If eligible, we will contact you to opt in or out.raised.

- Supporting Information uploaded as separate files, titled 'Dataset', 'Figure', 'Table', 'Text', 'Protocol', 'Audio', or 'Video'.

We hope to receive your revised manuscript within the next 30 days. If you anticipate any delay in its return, we ask that you let us know the expected resubmission date by email at ploscompbiol@plos.org.

Sincerely,

Florian Markowetz

Associate Editor

PLOS Computational Biology

Feilim Mac Gabhann

Editor-in-Chief

PLOS Computational Biology

[LINK]

Reviewer's Responses to Questions

**Comments to the Authors:**

Reviewer #1: Review: Hydra: A mixture modelling framework for subtyping pediatric cancer cohorts using multimodal gene expression by Pfeil J et al.

Purpose of the research: Develop an improved computational platform to molecularly classify rare pediatric cancers with high accuracy by transcriptomic data that can be implemented in clinical settings.

Problem: Current modalities are insufficient to accurately identify novel, subtle cancer specific precision oncogenic targets and networks above thresholds relating to tumour heterogeneity from the microenvironment and the cell-of-origin.

What are Pfeil J et al trying to accomplish: Create a highly accurate computational analysis package that can be applied to single samples and easily utilized in clinical settings to facilitate identifying better treatment options for rare pediatric cancers.

How is it done today: The authors highlight several of the numerous current approaches for analysing bulk RNAseq data from patient samples and their current limitations. Importantly, they compared their new package to two of the current standards: single-sample gene set enrichment anlaysis (ssGSEA) and gene set variation analysis (GSVA). Based on the defined metrics, their Hydra approach outperformed the other packages overall.

What’s new in their approach: The authors elicit a Dirichlet process mixture model to detect multimodally distributed genes and signatures. By subsetting the data via this method, much of the background noise from the tumour microenvironment heterogeneity is reduced. The authors highlight that their approach was previously not feasible computationally for clinical setting but with new developments in Bayesian variational inference it is now possible. Their approach specifically performs better when effect sizes are smaller and fewer genes are differentially expressed which is an ideal platform for rare tumour types such as pediatric cancers.

Overall impression/suggestions:

- The authors were very thorough in the explanations and rationale for the methodology which is extremely helpful for individuals with limited knowledge in computational biology. The rationale for the study is sound and the approach is very reasonable.

- The abstract and author summary are concise, clear and accurately reflect the data presented in the manuscript.

- Would the authors elaborate as to why they chose TCGA GBM data to test hydra on initially? This a tumour predominantly found in adults and that has higher mutational frequency than pediatric brain tumours.

- The authors analysed 5 pediatric tumour datasets from UCSC treehouse compendium. It is very good to see that hydra and its different components can be applied to all the tumour types. However, it was confusing when performing some analyses on only 1 type of cancer or a few and not others. For continuity, would the authors consider focusing on neuroblastoma for each of the methods presented and use the other cancer types as a more supporting role? E.g. hydra filter/enrich also on neuroblastoma data.

- Have the authors considered applying hydra on the more common pediatric tumours such as lymphoma, leukemia or brain tumours? For example medulloblastoma has very well defined molecular subtypes based on transcriptomic data (methylomic, proteomic, et..)…would hydra better refine these currently well-defined subgroups by Paul Northcott, Michael Taylor, Stefan Pfister, Richard Gilbertson, etc…

- The histological data (images and analysis) are underwhelming to support their models. For example, figure 3C would be much better supported if the authors had access to human patient samples to compare the findings of the xCell score based on hydra clusters. Immunohistochemistry for B-cells, CD8+ T-cells and fibroblasts are easy to do if samples are available. For the immune system, the spatial context is especially important when determining the ‘hot’ and ‘cold’ nature of the tumours.

- The N-of-1 tumour analysis is very attractive. Rates of pediatric cancers are low which is a hurdle in facilitating better treatments for those affected. Having a more rigid histological analysis of the samples used (figure 4) is needed. General lymphocyte content is in not sufficient for inflammation. The pathologist could supply additional information on neutrophil, macrophages, etc… to better describe the type of inflammation occurring. This level of pathological resolution is needed to better support the findings from Hydra to be utilized on n-of-1 findings.

- The hydra filter/enrich analysis of the osteosarcoma data is very interesting. I think these data would be better suited in the main figures not supplemental.

- For actionable drug targets, the authors highlight JAK1 signaling in Ewing Sarcoma. The authors highlight how important interpretation of the data are (Mast cell infiltration vs. the tumour) for selecting drug targets regarding the complexities of the tumour microenvironment. Can the authors supply a more concrete cancer specific actionable target to support the filter method of Hydra…something less nuanced?

- For S4B and C, these data are impressive and should be moved to main figures.

- Why is S7 in the supplemental data? Did the authors assess all cluster combinations and identify only wound healing and translational regulation as important in osteosarcoma…what about cell cycle,etc..

General comments:

- Line 22: Though many accept that pediatric tumours have fewer mutations relative to adult cancers, this is not true for all pediatric cancers. Additionally, the spectrum of tumours in paediatrics and adults is remarkably different. Would the authors supply some citation for their statement or provide a comparison on mutational frequencies…even focusing on brain tumours comparing peidatrics to adults. Stefan Pfister’s group in Heidelberg have published some data related to this.

- It may be my version, but the figures were very blurry and hard to make out details.

- Line 473: these findings are not too surprisingly based since the 5 cancers assessed have links back to neural crest cells.

- Line 482: should reference S7 not S6.

- The method section describing the histological analysis is not sufficient. Please see above for suggestions on what to highlight.

- Several graphs in the figures have no labelled axes.

Reviewer #2: In the manuscript “Hydra: A mixture modeling framework for subtyping pediatric cancer cohorts using multimodal gene expression signatures“, the authors proposed a novel method for classifying tumor samples based on their expression profiles. The method focuses on multimodally expressed genes (identified through a Dirichlet process mixture model) and its effects are demonstrated in cohorts of pediatric cancers. The method is compared to a standard method in the field, gene set enrichment analysis (GSEA) and its equivalent for N-of-One type of analysis, single-sample GSEA. The method is a non-parametric mixture model and contain three tools named filter, enrich and sweep. The first result is obtained from synthetic gene expression data and demonstrate that hydra outperformed ssGSEA and GSVA (Gene set variation analysis). The second one identified 3 subclusters of the MYCN non-amplified neuroblastoma samples and relates these 3 subclusters to features of the tumor microenvironment. The third result illustrate the use of hydra to classify individual samples based on an annotated compendium of expression profiles, with five MYCN non-amplified neuroblastoma samples and the classification provided in the previous section. The fourth result section demonstrate the use of hydra’s functions for unsupervised pathway enrichment on the TARGET osteosarcoma dataset and on a Ewing sarcoma dataset. The final result shows how the hydra analysis on small blue round cell tumors clustered samples based on shared expression profiles and regardless of histology, while TumorMap clustering separates these different tumor types.

The method is well described and the results nicely illustrate what it can achieve, on cohorts that are both accessible and of interest to the authors of the manuscript. I have a set of minor comments and recommendation to try to improve the manuscript.

Comments:

1) I think a discussion of runtime is warranted, especially in comparison with the other methods, and an estimation of how runtime scales with the number of samples in the analysis.

2) In the first result with synthetic data, the data is modelled “as a multivariate Gaussian distribution” (l.204), which is the running assumption in Hydra (l. 94). It would be more powerful to model the synthetic data with different assumptions and measure the performance of the method.

3) l.125: The URL www.dockerhub.com/jpfeil/hydra failed for me, but the correct URL seems to be https://hub.docker.com/r/jpfeil/hydra. I note that the title of the manuscript indicate that hydra is a method for “subtyping pediatric cancer cohorts”, but the short description of the Docker package describes it as a “clustering pipeline to identify differentially expressed pathways”. I get how those two things are related, but I believe it would be good to clarify in the text .

4) Some sections are very technical. For instance, the lines 111 to 116 are referring to concepts and methods that may not be common knowledge for part of the audience. I leave it to the authors to decide if they can explain it more or prefer to leave it there, assuming that the reader will consult the references cited.

5) I would appreciate a guide, on the Github page or even better as part of the manuscript, describing how to use the method on a completely novel cancer type and novel datasets. How to process the data, build the model and run the different tools of Hydra? Some documentation already exists, but could be expanded a bit more if the authors want more users to adopt their method.

**Have all data underlying the figures and results presented in the manuscript been provided?**

Reviewer #1: Yes

Reviewer #2: Yes

PLOS authors have the option to publish the peer review history of their article (what does this mean?). If published, this will include your full peer review and any attached files.

Reviewer #1: No

Reviewer #2: No

---

## [Decision Letter · Decision Letter 1]

28 Feb 2020

Dear Mr. Pfeil,

We are pleased to inform you that your manuscript 'Hydra: A mixture modeling framework for subtyping pediatric cancer cohorts using multimodal gene expression signatures' has been provisionally accepted for publication in PLOS Computational Biology.

Best regards,

Florian Markowetz

Deputy Editor

PLOS Computational Biology

Feilim Mac Gabhann

Editor-in-Chief

PLOS Computational Biology

Reviewer's Responses to Questions

Comments to the Authors:

Please note here if the review is uploaded as an attachment.

Reviewer #1: Thank you for addressing all the comments/concerns raised during the initial review.

The comments to the questions and alterations to the manuscript have satisfied this reviewer.

Reviewer #2: Thanks for taking into consideration my comments and for improving your manuscript. I recommend its publication and believe it will be well received by the scientific community

Have all data underlying the figures and results presented in the manuscript been provided?

Large-scale datasets should be made available via a public repository as described in the 

PLOS Computational Biology

data availability policy, and numerical data that underlies graphs or summary statistics should be provided in spreadsheet form as supporting information.

Reviewer #1: Yes

Reviewer #2: Yes

PLOS authors have the option to publish the peer review history of their article (what does this mean?). If published, this will include your full peer review and any attached files.

Do you want your identity to be public for this peer review?

 For information about this choice, including consent withdrawal, please see our Privacy Policy.

Reviewer #1: Yes: Eric P Rahrmann

Reviewer #2: No

---

## [Editor Report · Acceptance letter]

1 Apr 2020

PCOMPBIOL-D-19-01924R1 

Hydra: A mixture modeling framework for subtyping pediatric cancer cohorts using multimodal gene expression signatures

Dear Dr Pfeil,

I am pleased to inform you that your manuscript has been formally accepted for publication in PLOS Computational Biology. Your manuscript is now with our production department and you will be notified of the publication date in due course.

With kind regards,

Laura Mallard
